# Locating the Mandibular Lingula Using Cone-Beam Computed Tomography: A Literature Review

**DOI:** 10.3390/jcm12030881

**Published:** 2023-01-22

**Authors:** Chun-Ming Chen, Hui-Na Lee, Ying-Ting Chen, Kun-Jung Hsu

**Affiliations:** 1School of Dentistry, College of Dental Medicine, Kaohsiung Medical University, Kaohsiung 80708, Taiwan; 2Division of Oral and Maxillofacial Surgery, Department of Dentistry, Kaohsiung Medical University Hospital, Kaohsiung 80756, Taiwan; 3Division of Conservative Dentistry, Kaohsiung Medical University Hospital, Kaohsiung 80756, Taiwan; 4School of Dentistry, CEU Cardenal Herrera University, 46113 Valencia, Spain; 5Department of Dentistry, Kaohsiung Medical University Hospital, Kaohsiung 80756, Taiwan

**Keywords:** mandibular lingula, mandibular foramen, inferior alveolar nerve block, occlusal plane, cone-beam computed tomography, ramus surgery

## Abstract

This study aimed to review the literature on adult mandibular lingula (ML) locations and related distances determined using cone-beam computed tomography (CBCT). A search was conducted for studies on CBCT using the following databases: PubMed, Web of Science, and Embase. The search results were limited to studies published between 1970 and 2021. The inclusion criteria were the investigation of ML location, CBCT, and participants aged ≥18 years. Eligible studies were examined for the distances from the lingual tip to the anterior ramus border, posterior ramus border, sigmoid notch, inferior ramus border, and occlusal plane. Eight studies on CBCT qualified for inclusion in the study. The mean distances from the ML to the anterior ramus border were 15.57 to 20 mm. In most of these, the ML was located above the occlusal plane. No significant differences were observed in the location and related distances for the ML among patients of different sexes, ethnicities, or skeletal patterns.

## 1. Introduction

Identifying the position of the mandibular foramen (MF) is essential for the successful administration of an inferior alveolar nerve block (IANB) and for avoiding damage to the inferior alveolar neurovascular bundle during ramus surgery. Located on the medial surface of the ramus, the MF is a bony orifice through which the inferior alveolar neurovascular bundle enters the mandible. The mandibular lingula (ML), located above the MF, is a key anatomical landmark for determining the position of the MF and for the orientation of the surgeon during ramus surgery. The ML is a bone plate structure that medially limits the MF and provides an attachment area for the sphenomandibular ligament [1]. Determining the shape, position, and height of the ML is crucial for dentists and surgeons during IANBs and ramus surgeries [2]. Understanding the position of the ML and the surrounding nerves and blood vessels is critical for a successful operation, thereby preventing excessive intraoperative blood loss and postoperative numbness in the lower lip [3,4].

In the past, determination of the morphometry of the lingula required using the dry mandible [5,6,7,8,9,10]. Recently, cone-beam computed tomography (CBCT) has provided high quality and accurate three-dimensional images to replace dry mandibles in morphometric study [11,12,13,14]. Studies [15,16,17,18,19,20,21,22] have employed CBCT images to determine the shape of the ML and measure the distances related to the ML. Although CBCT provides real-time measurements, the results obtained using the images may be affected by the resolution used for image processing and the applied image-recognition method. Furthermore, the ramus borders to ML distances are closely related to the horizontal and vertical reference planes set for CBCT scans and are considerably influenced by the consistency of the reference point. Therefore, this study reviewed articles on CBCT scans separately and explored the differences in the relative position of the ML on the ramus and its distance from other anatomical landmarks among patients of different sexes, ethnicities, and skeletal patterns. The present study included all reference planes used in the articles on CBCT.

## 2. Materials and Methods

### 2.1. Search Strategy

A literature search was conducted using the PubMed, Web of Science, and Embase databases. The search results were limited to studies published between 1970 and 2021. Results of participants aged <18 years were excluded from the review. The first search term used was “mandibular lingula location.” These results were further examined for the inclusion of CBCT. 

### 2.2. Study Selection and Eligibility

Two authors of this study selected the articles using a data retrieval system. After reading the titles and abstracts, the authors independently assessed the selected articles based on eligibility criteria. The inclusion criteria were as follows: (1) Randomized controlled trials and observational studies; (2) Research participants aged ≥18 years. The exclusion criteria were as follows: (1) Non-English articles; (2) Non-human studies; (3) Duplicate articles. The authors read the full text of any article for which they could not reach a consensus regarding eligibility. Regarding the inter-rater reliability, the kappa coefficient was 0.890 (*p* < 0.05), demonstrating a high level of consistency between the two authors’ discretionary eligibility selections.

### 2.3. Data Extraction and Analysis of ML Location

The data were examined independently by the two authors according to the research methodology. ML was identified as a reference landmark, and its related distances were obtained. If the results extracted by the two authors from the same article were inconsistent, the authors discussed the article with other authors to reach a consensus. The ML location was determined based on relevant reference planes (Figure 1). The ML-related distances (Figure 2) were measured using the following landmarks: lingula tip to anterior ramus border (As: landmark (shortest distance) at the anterior border of ramus; Ai: intersection landmark parallel to horizontal plane at the anterior border of ramus), posterior ramus border (Ps: landmark (shortest distance) at the posterior border of ramus; Pi: intersection landmark parallel to the horizontal plane at the posterior border of ramus), sigmoid notch (Ss: lowest location (shortest distance) of the sigmoid notch; Si: intersection landmark parallel to vertical plane and through lingula at the sigmoid notch), inferior ramus border (Is: landmark (shortest distance) at the inferior border of ramus; Ii: intersection landmark parallel to vertical plane and through lingula at the inferior border of ramus), and occlusal plane.

## 3. Results

### 3.1. Study Selection and Eligibility

Using the term “mandibular lingula location” (Figure 3), 121 articles were identified from electronic databases (PubMed, *n* = 53; Web of Science, *n* = 36; Embase, *n* = 32). Regarding the mandibular lingual location and CBCT, 29 articles were identified from electronic databases (PubMed, *n* = 12; Web of Science, *n* = 11; Embase, *n* = 6). After verifying the eligibility of the selected articles, eight remained for inclusion in the study and full-text review by the authors.

### 3.2. Data Extraction and Analysis

#### 3.2.1. ML Height

The sample size of the CBCT group ranged from 60 to 412 participants (Table 1); the youngest participant was 18 years old and the oldest participant was 71 years old. The occlusal plane was used as the coordinate axis for measurements in five studies [16,18,19,20,22]. The mean ML height ranged from 7 to 9.08 mm [16,17,21].

#### 3.2.2. Distances from Lingual Tip to Anterior Ramus Border and Posterior Ramus Border 

The mean distance of the ML to the anterior ramus border ranged from 15.57 to 20 mm [15,16,17,18,19,20,21,22]. The mean distance of the ML to the posterior ramus border ranged from 12 to 18.2 mm [15,16,17,18,19,20,21,22]. 

#### 3.2.3. Distances from Lingual Tip to Sigmoid Notch and Inferior Ramus Border

The mean distance of the ML to the sigmoid notch ranged from 13.34 to 20.72 mm [15,16,17,18,19,20,21,22]. The mean distance of the ML to the inferior ramus border ranged from 30.4 to 39.1 mm [15,16,17,18,19,20,21,22].

#### 3.2.4. Distances from Lingual Tip to Occlusal Plane

Most ML locations were above the occlusal plane. Five articles reported that the ML was above the occlusal plane, with a distance range of 3.57 to 11.77 mm [16,18,19,20,22]. 

## 4. Discussion

Mandibular growth is fully developed at approximately 18 years of age. Therefore, studies were excluded from our review if the included patients were younger than 18. The outcomes of the eligible studies for different sexes, races, and skeletal patterns are discussed below.

### 4.1. ML Height

Sekerci and Sisman [16] observed that the ML height on the right side of the mandible was greater in men than in women. Hsu et al. [21] reported that the ML height was significantly greater in men than in women and that there were no significant differences in the ML height between the left and right sides of the mandible. They also noted no significant differences in ML height among patients with Class I, Class II, or Class III malocclusion. ML height had similar dimensions in different races and skeletal patterns.

### 4.2. Anterior Ramus Border to Mandibular Lingula Distance (ARL Distance)

The reported distance of the ML from the anterior ramus border differed considerably between studies. Findik et al. [15] reported a mean distance of 15–16 mm, whereas Hsu et al. [21] reported a distance of 18–20 mm. These inconsistencies arose because each study used different reference points and planes (horizontal and vertical). Findik et al. [15] directly measured the ML distance from the anterior ramus border using the axial plane as the reference plane, thereby obtaining the lowest measurement results. Other studies [16,18,20] that used the occlusal plane as the reference plane obtained measurement results larger than those of Findik et al. [15] but smaller than those of Hsu et al. [21], who measured the distance using the Frankfort horizontal plane as the reference plane. 

Several studies [17,18,21,22] reported no significant differences between the sexes or between the left and right sides of the mandible. However, Findik et al. [15] reported that the distance between the ML and the anterior ramus border on the left side of the mandible was significantly greater than that on the right side. Hsu et al. [21] reported that the distance of the ML from the anterior ramus border was significantly greater in men than in women but observed no significant differences in the distance of the ML from the anterior ramus border in patients with different skeletal patterns. However, Zhou et al. [18] reported that the ML distance from the anterior ramus border was significantly greater in men with a low gonial angle (LGA) than that in men with a high gonial angle (HGA). Among the distances to the lingual, the anterior ramus border to the lingula distance is more important than the posterior border to the lingual distance.

For clinical application, we recommend using the occlusal plane as a reference plane. The measured landmark is a plane running parallel to the occlusal plane, which runs through the ML and intercepts the anterior ramus border. This method should result in more accurate administration of IANB and avoid damage to the inferior alveolar neurovascular bundle during ramus surgery.

### 4.3. Posterior Ramus Border to Lingula Distance

Some studies [16,18,19,20,21] stated that the distance between the ML and the posterior ramus border was significantly greater in men than in women, except for Lupi et al. [22], who observed no significant differences between the sexes. Sekerci and Sisman [16] proposed that the distance between the ML and the posterior ramus border is significantly greater on the left side of the mandible in men than that on the left side of the mandible in women. Seneal et al. [17] and Hsu et al. [21] noted no significant differences in the distance between the ML and the posterior ramus border on the left and right sides of the mandible. Hsu et al. [21] also observed no significant differences between participants with different classes of malocclusions. However, Zhou et al. [20] reported that the distance between the ML and the posterior ramus border was significantly greater in men with LGA than that in men with HGA, in women with LGA than that in women with HGA, in men with LGA than that in women with LGA, and in men with HGA than that in women with HGA. 

Participants of different races, ages, and sexes were recruited in each of the aforementioned studies [15,16,17,18,19,20,21,22]. The use of different reference planes, including the occlusal plane, Frankfort horizontal plane, and axial plane, resulted in considerable differences in the measured distances. Regarding the various landmarks at the posterior ramus border, we suggest that the occlusal plane is a better reference plane to set both landmarks of anterior and posterior ramus borders. 

### 4.4. Sigmoid Notch to Lingual Distance

Regarding the distance from the ML to the sigmoid notch, Lupi et al. [22] reported that the lowest mean distance was 13.34 mm in women. However, the measurements were not performed using the lowest point of the sigmoid notch. Instead, the authors measured the distance vertical to the occlusal plane from the lingula tip to the sigmoid notch, thereby obtaining a lower value. Lupi et al. [22] and Hsu et al. [21] reported no significant differences between sexes. Seneal et al. [17] and Hsu et al. [21] observed no significant difference between the left and right sides of the mandible. Hsu et al. [21] reported no significant differences in the distance between the ML and sigmoid notches in patients with different skeletal patterns. However, Akcay et al. [19] noted that the distance between the ML and the sigmoid notch was significantly greater in men with Class III malocclusion than that in women with Class III malocclusion. However, there were no significant differences in the distance of the ML from the sigmoid notch in men and women with Class I malocclusion. Zhou et al. [18] observed that the distance between the ML and the sigmoid notch was significantly greater in men with LGA than in those with HGA. Moreover, the number of men with LGA and HGA was greater than the number of women with LGA and HGA. For consistency, we recommend that the measured landmark be the lowest point of the sigmoid notch. 

### 4.5. Inferior Ramus Border to Lingula Distance

Among all studies, Senel et al. [17] reported the greatest distance (38.3 mm) of the ML from the inferior ramus border. This result can be attributed to the reference plane used by Senel et al. [17]. Several articles [16,18,20,21] reported that the distance between the ML and the inferior ramus border was significantly greater in men than in women. However, Lupi et al. [22] reported no significant differences in the distance between the ML and the inferior ramus border between the sexes. Seneal et al. [17] and Hsu et al. [21] noted no significant differences in the ML distance from the inferior ramus border between the left and right sides of the mandible. Hsu et al. [21] observed no significant differences in the distance between the ML and the inferior ramus border in patients with different classes of malocclusion. However, Zhou et al. [18] reported that the distance between the ML and the inferior ramus border was significantly greater in men with LGA than that in men with HGA, and in women with LGA than that in women with HGA. Moreover, the number of men with LGA and HGA was greater than the number of women with LGA and HGA. In each of the aforementioned studies, participants of different ethnicities, ages, and sexes were recruited. Different reference planes, sigmoid notch landmarks, and inferior border landmarks were used, resulting in considerable differences in the measured distances. Tracing an obvious notch at the inferior border of the mandible is not always possible. Therefore, we recommend identifying two landmarks: (1) Setting a plane through the ML vertically to the occlusal plane and intercepting the inferior border; (2) Setting a plane from the lowest point of the sigmoid notch through to the inferior border.

### 4.6. Lingula to Occlusal Plane Distance

Akcay et al. [19] reported the distance of the ML from the occlusal plane as 8–10 mm, whereas Lupi et al. [22] reported this distance as 10–12 mm. There were no significant differences in the distance of the ML from the occlusal plane between the sexes or between the left and right sides of the mandible. Akcay et al. [19] reported that the distance between the ML and the occlusal plane was significantly greater in patients with Class III malocclusion than that in those with Class I malocclusion. Zhou et al. [18] reported that this distance was higher in men with LGA than that in women with LGA, and in men with HGA than that in women with HGA. Lingula-to-occlusal plane distance is more important than other distances in terms of administering an IANB and performing ramus surgery. CBCT is a useful tool for identifying ML morphology, location, and relation to the occlusal plane to avoid damage to the neurovascular bundle.

### 4.7. Clinical Relevance and Limitations of the Study

Numerous studies [15,16,17,18,19,20,21,22,23,24,25,26] have been conducted on the position of the ML and have presented different results. CBCT images were affected by the resolution used for image processing. Moreover, the varied settings for different reference planes and points may have influenced the results of the study. Regarding clinical relevance, the occlusal plane is the preferred reference plane to identify the ML and MF to prevent injury to the inferior alveolar neurovascular bundle.

The limitations of this literature review were as follows: (1) Varied reference planes; (2) Rare reports of edentulous people; (3) Imbalance in the age distribution of the population; (4) Racial distribution disequilibrium of the population. For clinical applications, we recommend using the occlusal plane as the reference plane in future studies. However, there should be modifications to the occlusal plane if the patient is edentulous. Furthermore, future studies should include a sample distribution that considers the age and race of the study population.

## 5. Conclusions

Differences in age, race, and sex of participants, and in reference planes and points may have generated different measurement results. Significant differences were observed in the ML height between patients of different sexes and races and of those with different skeletal patterns. In most studies, the ML was located above the occlusal plane. In addition, most studies reported no significant differences in the distance of the ML to the anterior ramus border, posterior ramus border, sigmoid notch, or inferior ramus border among patients of different sexes and races and among those with different skeletal patterns. Additionally, the distances of the ML to the anterior ramus border, posterior ramus border, sigmoid notch, and inferior ramus border were significantly greater in patients with LGA than those in patients with HGA. 

## Figures and Tables

**Figure 1 jcm-12-00881-f001:**
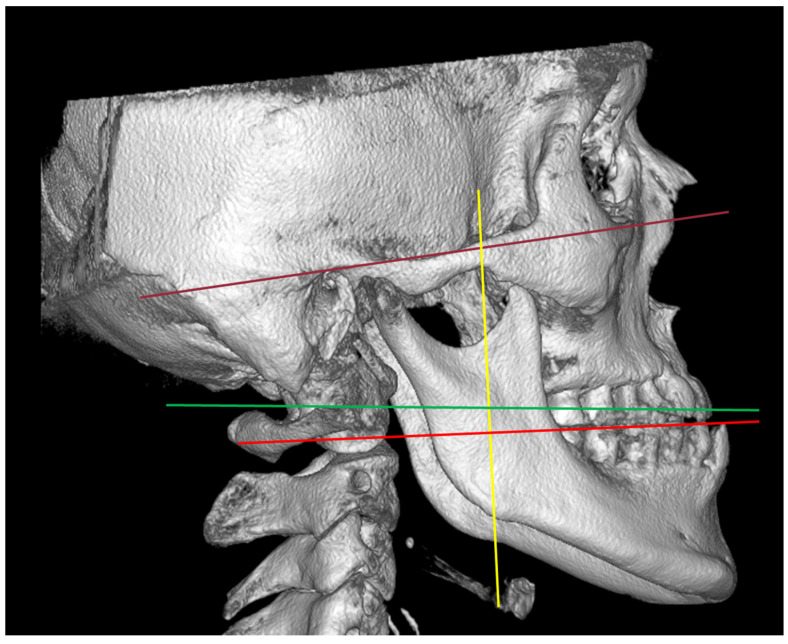
Reference lines: Brown line: Frankfort horizontal plane; Green line: axial plane of ramus; Red line: occlusal plane; Yellow line: vertical plane to occlusal plane.

**Figure 2 jcm-12-00881-f002:**
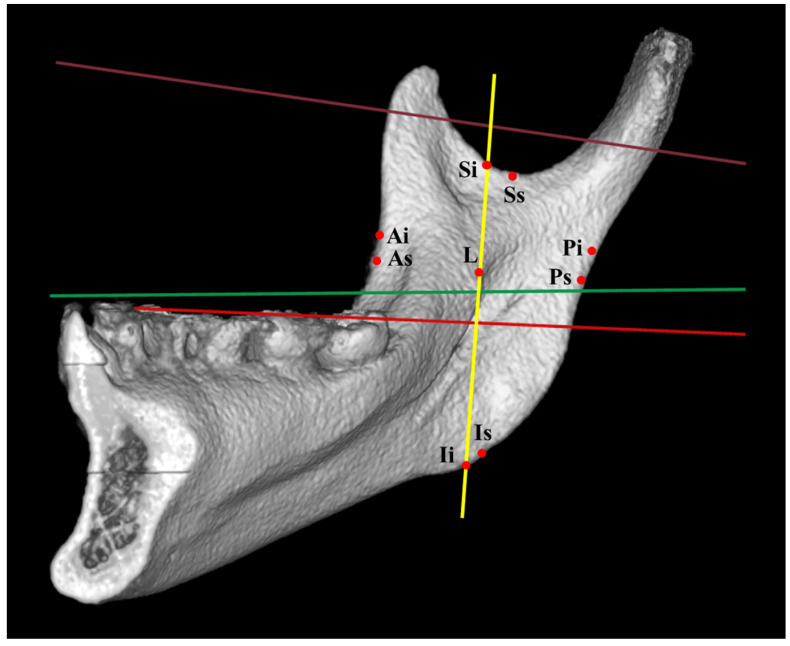
Reference lines: Brown line: Frankfort horizontal plane; Green line: axial plane of ramus; Red line: occlusal plane; Yellow line: vertical plane to occlusal plane. Landmarks: L: lingual, As: landmark (shortest distance) at the anterior border of ramus; Ai: intersection landmark parallel to horizontal plane at the anterior border of ramus; Ps: landmark (shortest distance) at the posterior border of ramus; Pi: intersection landmark parallel to horizontal plane at the posterior border of ramus; Ss: deepest location (shortest distance) of the sigmoid notch; Si: intersection landmark parallel to vertical plane and through lingula at the sigmoid notch; Is: landmark (shortest distance) at the inferior border of ramus; Ii: intersection landmark parallel to vertical plane and through lingula at the inferior border of ramus.

**Figure 3 jcm-12-00881-f003:**
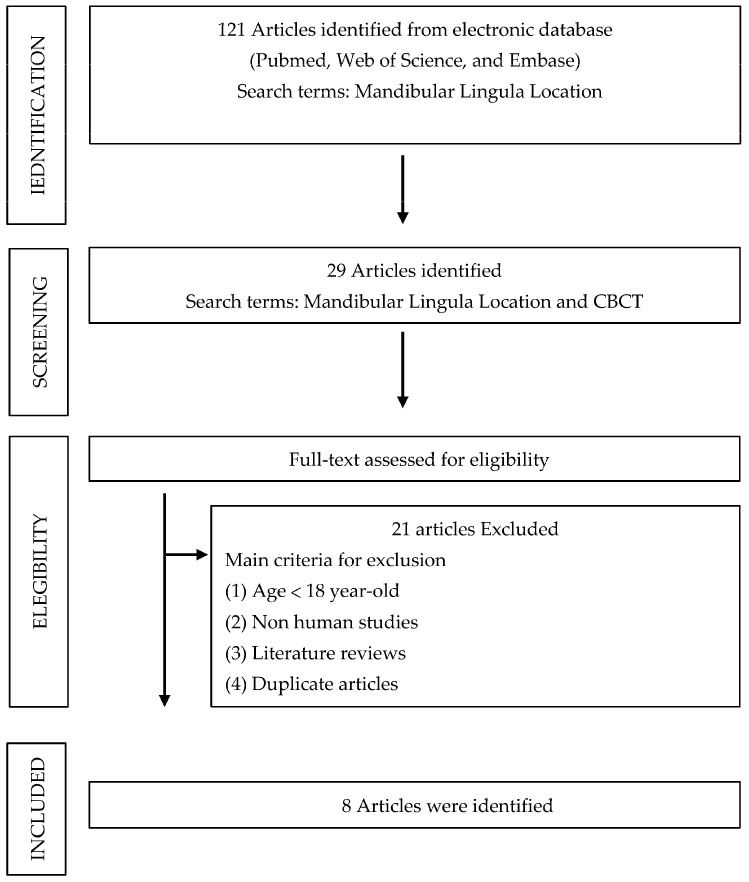
Process flow of article selection in the mandibular lingula location using cone-beam computed tomography.

**Table 1 jcm-12-00881-t001:** Demographic and study characteristics in the included studies (cone-beam computed tomography: CBCT).

Author	Age/Range (Years)	Patients/Total Sides	Axis of Coordinates	Sex	Race	Landmarks	Height of Lingula	Anterior Ramus to Lingula	Posterior Ramus to Lingula	Sigmoid Notch to Lingula	Inferior Border to Lingula	Above Occlusal Plane	Below Occlusal Plane
Year				F (Female)		Right (R)							
Country of Origin				M (Male)		Left (L)							
Findik et al. [15]	19–71 years	*n* = 96	Axial plane	NA	Turk	Ai	NA	15.57 (R)	NA	NA	NA	NA	NA
2014		192 side						15.73 (L) *	NA	NA	NA	NA	NA
Turkey													
Sekerci and Sisman [16]	Adult	*n* = 412	Occlusal plane	Total (199 F, 213 M)	Turk	Ai, Pi, Ss, Ii	7.91	16.77	13.02	15.32	33.43	3.6	NA
2014		824 side		F		R	7	15.97	12.43	13.95	30.93	3.62	NA
Turkey				M		R	9.08 *	18.23 *	13.61	17.21 *	35.53 *	3.66	NA
				F		L	7.24	15.6	12	14.17	30.94	3.57	NA
				M		L	8.58	17.26	14.03 *	15.93	36.32 *	3.58	NA
Senel et al. [17]	46	*n* = 63	Axial plane	28 F, 35 M	Turk	Ai, Pi, Ss, Ii							
2015	25–70 years	126 sides				Total	7.8	18.5	16.9	18.1	38.3	NA	NA
Turkey						R	8.3	18.4	16.7	18.1	37.6	NA	NA
						L	7.4	18.7	17.1	18.1	39.1	NA	NA
Zhou et al. [18]	26.8	*n* = 106	Occlusal plane	55 F	Korean	Ai, Pi, Ss, Ii	NA	18.3	17	15.5	30.5	5.8	1.9
2017	18–36 years	121 sides		51 M			NA	18.2	18.2 *	15.7	35.3 *	6.2	0
Korea													
Akcay et al. [19]	Class I	*n* = 30	Occlusal plane	Total	Turk	As, Ps, Ss	NA	NA	16.55	17.7	NA	8.12	NA
2019	18–37 years	60 sides		14 F			NA	NA	15.92	17.07	NA	7.60	NA
Turkey				16 M			NA	NA	17.09 *	18.24	NA	8.58	NA
	Class III	*n* = 30		Total			NA	NA	15.82	18.73	NA	9.91 *	NA
	18–36 years	60 sides		16 F			NA	NA	15	17.81	NA	9.89	NA
				14 M			NA	NA	16.75 *	19.79 *	NA	9.93	NA
Zhao et al. [20]		*n* = 407	Occlusal plane	206 F, HGA (270 sides)	Chinese	As, Ps, Ss, Is	NA	16.77	16.42	16.22	32.37	5.03	NA
2019	20–35 years	814 sides		LGA (142 sides)			NA	17.18	17.05 *	16.6	33.32 *	5.21	NA
China				201 M, HGA (218 sides)			NA	16.53	16.9	16.76	34.74	5.97	NA
				LGA (184 sides)			NA	17.63 *	17.94 *	17.64 *	36.86 *	5.88	NA
				HGA (F = 270 sides)			NA	16.77	16.42	16.22	32.37	5.03	NA
				(M = 218 sides)			NA	16.53	16.9 *	16.76 *	34.74 *	5.97 *	NA
				LGA (F = 142 sides)			NA	17.18	17.05	16.6	33.32	5.21	NA
				(M = 184 sides)			NA	17.63	17.94 *	17.64 *	36.86 *	5.88 *	NA
Hsu et al. [21]	Adult	*n* = 72	Frankfort	Total	Taiwanese	Ai, Pi, Ss, Ii	8.07	19.21	15.22	20.04	31.2	NA	NA
2020		144 sides	horizontal	49 F			7.76	18.85	14.89	19.99	30.4	NA	NA
Taiwan			plane	23 M			8.73 *	19.99 *	15.93 *	20.14	32.91 *	NA	NA
						R	8.2	19.23	15.25	20.51	31.26	NA	NA
						L	7.95	19.2	15.19	19.57	31.14	NA	NA
						Class I (*n* = 26)	8.24	18.86	15.36	19.74	31.28	NA	NA
						Class II (*n* = 21)	7.83	18.71	15.51	20.72	31.4	NA	NA
						Class III (*n* = 25)	8.11	20	14.83	19.77	30.95	NA	NA
Lupi et al. [22]	34.93	*n* = 111	Occlusal plane	Total	Italian	Ai, Pi, Si, Ii	NA	16.96	15.28	13.87	31.2	11.22	NA
2021	18–88 years	201 sides		43 F			NA	16.9	14.85	13.34	30.05	11.77	NA
Italy				68 M			NA	17.05	16.04	15.01	33.06	10.87	NA

*n*: number of patients; NA: not available; *: statistically significant; Lingula tip to anterior ramus border. As: landmark (shortest distance) at the anterior border of ramus, Ai: intersection landmark parallel to horizontal plane at the anterior border of ramus; Lingula tip to posterior ramus border. Ps: landmark (shortest distance) at the posterior border of ramus, Pi: intersection landmark parallel to horizontal plane at the posterior border of ramus; Lingula tip to sigmoid notch. Ss: lowest location (shortest distance) of the sigmoid notch, Si: intersection landmark parallel to vertical plane and through lingula at the sigmoid notch; Lingula tip to inferior ramus border. Is: landmark (shortest distance) at the inferior border of ramus, Ii: intersection landmark parallel to vertical plane and through lingula at the inferior border of ramus.

## Data Availability

Not applicable.

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
