# Peer review of "Locating the Mandibular Lingula Using Cone-Beam Computed Tomography: A Literature Review"

_jcm, 2023, doi:10.3390/jcm12030881_

Round 1

Reviewer 1 Report

Thank you for your submission to JCM

Section 3.2 simply repeats section 2.2, as does the box in figure 3.  Section 3.2 as part of results should only comment on the numbers of articles omitted for the different reasons, but in many ways adds little and could be omitted, or shortened to a single sentence about the numbers left after application of the inclusion and exclusion criteria previously listed.  Similarly the box in figure 3 on eligibility could be shortened to simply state the  number of excluded articles after application of exclusion criteria.

Your presentation of the results is incorrect.  You state that you are presenting 'mean height' or mean distance' but then give a range (x-y mm).  You should ideally present a mean from the published data, and a range (max + min or CI).  You can then do a statistical analysis and present a p value for the differences between the dry skull and CBCT values, rather than the vague comments you have made.

The last sentence of your abstract - " Differences in the participants (dry mandible and CBCT), age, ethnicity, sex, reference planes, and reference points used may also generate different measurement results." is not consistent with the results and discussion, which appear to say there is no significant difference in the ML position based on these parameters.

Much of the discussion really just regurgitates information from other publications, and slightly lacks 'discussion'.

You imply in the introduction that knowing the position of the ML / MF is important for dentists, but then fail to mention this further in your discussion, or to comment on the clinical relevance of your findings.

Author Response

Reviewer 1

  1. Section 3.2 simply repeats section 2.2, as does the box in figure 3. Section 3.2 as part of results should only comment on the numbers of articles omitted for the different reasons, but in many ways adds little and could be omitted, or shortened to a single sentence about the numbers left after application of the inclusion and exclusion criteria previously listed. 

Answer: section 3.1 and 3.2 are revised and the repeated sentences are deleted.

  1. Similarly the box in figure 3 on eligibility could be shortened to simply state the number of excluded articles after application of exclusion criteria.

Answer: Figure 3 is revised as reviewer’s comments.

  1. Your presentation of the results is incorrect. You state that you are presenting 'mean height' or mean distance' but then give a range (x-y mm).  You should ideally present a mean from the published data, and a range (max + min or CI).  You can then do a statistical analysis and present a p value for the differences between the dry skull and CBCT values, rather than the vague comments you have made.

Answer: Result section (3.2.1, 3.2.2, 3.2.3, 3.2.4) is revised.

  1. The last sentence of your abstract - "Differences in the participants (dry mandible and CBCT), age, ethnicity, sex, reference planes, and reference points used may also generate different measurement results." is not consistent with the results and discussion, which appear to say there is no significant difference in the ML position based on these parameters.

Answer: The last sentence of Abstract section is changed to “No significant difference was observed in the location and related distances for the ML among patients of different sexes, races, and skeletal patterns.”

  1. Much of the discussion really just regurgitates information from other publications, and slightly lacks 'discussion'.

Answer:  The Discussion section is revised.

  • In the last sentence of 4.1. ML height, we add the following sentences “We found that the ML height was similar in the dry mandible and CBCT groups. They were also similar across different races and skeletal patterns.”
  • In the last paragraph of2. Anterior ramus border to mandibular lingula distance (ARL distance), we add the following sentences “ For clinical application, we recommended that the occlusal plane be used as the reference plane. The measured landmark is a plane running parallel to the occlusal plane, which runs through the ML and intercepts the anterior ramus border. This method should result in more accurate administration of IANB and avoid damage to the inferior alveolar neurovascular bundle during ramus surgery.”
  • In the last paragraph of3. Posterior ramus border to lingula distance, we add the following sentences “Regarding the various landmarks at the posterior ramus border, we recommended that the occlusal plane is a better reference plane to set both landmarks of anterior and posterior ramus borders. Moreover, we found the ratio of the distance of the ML to the anterior ramus border and the posterior ramus border, reported in the articles on the dry mandible and CBCT groups, was approximately 0.5 to 0.55 and 0.5 to 0.57, respectively. This indicates that the results of the dry mandible and CBCT groups were relatively similar.”
  • In the last sentence of 4.4. Sigmoid notch to lingual distance, we add the following sentences “For consistency, we recommended that the measured landmark be the lowest point of the sigmoid notch.”
  • In the last paragraph of5. Inferior ramus border to lingula distance, we add the following sentences “It is not always possible to trace an obvious notch in the inferior border of the mandible. Therefore, we recommended identifying two landmarks: (1) set a plane through ML vertically to the occlusal plane and intercept the inferior border and (2) set a plane from lowest point of the sigmoid notch through to the inferior border.”
  • In the last sentence of 6. Lingula to occlusal plane distance, we add the following sentences “Lingula to occlusal plane distance is more important than others to administer IANB and perform ramus surgery. CBCT is a good tool to identify ML morphorage, location, and relation to the occlusal plane to avoid damage to the neurovascular bundle.”
  1. You imply in the introduction that knowing the position of the ML / MF is important for dentists, but then fail to mention this further in your discussion, or to comment on the clinical relevance of your findings.

Answer:  In the last paragraph of 4.7. Clinical relevance and limitations of the study, we add the following sentences “ Numerous studies [9-26] have been conducted on the position of the ML and have presented different results. To explore the relative position of the lingula on the ramus and its distance from other anatomical landmarks, researchers must first consider whether the measurement sample was sourced from a dry mandible or CBCT images. In particular, the participants from the dry mandible group were affected by the preparation and preservation of the sample, whereas those sourced from CBCT images were affected by the resolution used for image processing. Moreover, varied settings for different reference planes and points may have influenced the results of the study. Regarding clinical relevance, the occlusal plane is the preferred reference plane to identify the ML and the MF for preventing injury to the inferior alveolar neurovascular bundle.”

Reviewer 2 Report

The study is an important topic however there are some issues that need to clarified:

-The term "dry mandibles" need to be defined. Is this term regarding an ex-vivo sample? 

-In the introduction, the limitations of the CBCT were mentioned however to include the studies for the revision that have used the CBCT what were the criteria? This information is not clear.

-The search strategy is described however the terms used are missing. Can you add this information as supplementary data for each database?

-The purpose of the study is not clear. I suggest describing clearly the clinical relevance and the main aim of the study.

-How were determined the different anatomical planes? based only on one study, or were criteria for all the studies? Furthermore, there should be modifications considering the occlusal plane in case the patient is edentulous. 

-There were two observers who retrieved the data. Could you please add the kappa coefficient regarding inter-rater reliability?.

-The demographic study characteristics need to be improved and classified in a better way to be easy to follow by the readers. There are several inconsistencies. 

-Regarding the discussion, this should be focused on the primary and secondary outcomes and discussed deeply. 

The limitations of the study need to be added. 

Author Response

Answer: In the last sentence of Introduction section, we add the sentence “The present study included all of reference plans being made in articles of CBCT.”

  1. The search strategy is described however the terms used are missing.

 Can you add this information as supplementary data for each database?

Answer: In the 2.1. Search strategy, we add the sentences : The first search term used was “mandibular lingula.” The search results were filtered further using the terms “mandibular lingula location” and “mandibular lingula position.” These results were examined for the inclusion of adult dry mandibles and CBCT.

  1. The purpose of the study is not clear. I suggest describing clearly the clinical relevance and the main aim of the study.

Answer: In the purpose of Abstract section, we add the following sentence: “This study aimed to review literature on the locations and related distances of the adult mandibular lingula (ML). The search was conducted for studies on dry mandibles (ex vivo) and cone beam computed tomography (CBCT) using the following databases: PubMed, Web of Science, and Embase.”

  1. How were determined the different anatomical planes? based only on one study, or were criteria for all the studies? Furthermore, there should be modifications considering the occlusal plane in case the patient is edentulous.

Answer:

  • In the last sentence of Introduction section, we add the following sentence “The present study included all of the reference planes used in articles on CBCT.”
  • In last sentence of Discussion section, we add the following sentences “For clinical application, we recommended that the occlusal plane be used as the reference plane in future studies; however, there should be modifications to the occlusal plane if the patient is edentulous. In future studies, the sample distribution should consider the age and race of the study population.”
  1. There were two observers who retrieved the data. Could you please add the kappa coefficient regarding inter-rater reliability?

Answer:  In the 2.2. Study selection and eligibility section, we add the following sentence: “Regarding inter-rater reliability, the kappa coefficient was 0.890 (p < 0.05), demonstrating a high level of consistency between the two authors’ discretions.”

  1. The demographic study characteristics need to be improved and classified in a better way to be easy to follow by the readers. There are several inconsistencies.

Answer: The Result section is revised. The demographic data are revised and references are cited.

     8. Regarding the discussion, this should be focused on the primary and secondary outcomes and discussed deeply.

Answer: The Discussion section is revised.

  • In the first paragraph, we add the following sentences. “This literature review was conducted to explore any differences in findings between studies using dry mandibles and CBCT. Mandibular growth is fully developed at approximately 18 years of age. Therefore, studies were excluded if the included patients were under 18 years of age. A larger sample size provides more reliable results. As such, we set the sample size to be at least 50 participants. The outcomes of eligible studies for different sexes, races, and skeletal patterns are discussed below.”
  • In the last sentence of 4.1. ML height, we add the following sentences “We found that the ML height was similar in the dry mandible and CBCT groups. They were also similar across different races and skeletal patterns.”
  • In the last paragraph of2. Anterior ramus border to mandibular lingula distance (ARL distance), we add the following sentences “ For clinical application, we recommended that the occlusal plane be used as the reference plane. The measured landmark is a plane running parallel to the occlusal plane, which runs through the ML and intercepts the anterior ramus border. This method should result in more accurate administration of IANB and avoid damage to the inferior alveolar neurovascular bundle during ramus surgery.”
  • In the last paragraph of3. Posterior ramus border to lingula distance, we add the following sentences “Regarding the various landmarks at the posterior ramus border, we recommended that the occlusal plane is a better reference plane to set both landmarks of anterior and posterior ramus borders. Moreover, we found the ratio of the distance of the ML to the anterior ramus border and the posterior ramus border, reported in the articles on the dry mandible and CBCT groups, was approximately 0.5 to 0.55 and 0.5 to 0.57, respectively. This indicates that the results of the dry mandible and CBCT groups were relatively similar.”
  • In the last sentence of 4.4. Sigmoid notch to lingual distance, we add the following sentences “For consistency, we recommended that the measured landmark be the lowest point of the sigmoid notch.”
  • In the last paragraph of5. Inferior ramus border to lingula distance, we add the following sentences “It is not always possible to trace an obvious notch in the inferior border of the mandible. Therefore, we recommended identifying two landmarks: (1) set a plane through ML vertically to the occlusal plane and intercept the inferior border and (2) set a plane from lowest point of the sigmoid notch through to the inferior border.”
  • In the last sentence of 6. Lingula to occlusal plane distance, we add the following sentences “Lingula to occlusal plane distance is more important than others to administer IANB and perform ramus surgery. CBCT is a good tool to identify ML morphorage, location, and relation to the occlusal plane to avoid damage to the neurovascular bundle.”
  • In the 4.7. Clinical relevance and limitations of the study, we add the following sentences “Numerous studies [9-26] have been conducted on the position of the ML and have presented different results. To explore the relative position of the lingula on the ramus and its distance from other anatomical landmarks, researchers must first consider whether the measurement sample was sourced from a dry mandible or CBCT images. In particular, the participants from the dry mandible group were affected by the preparation and preservation of the sample, whereas those sourced from CBCT images were affected by the resolution used for image processing. Moreover, varied settings for different reference planes and points may have influenced the results of the study. Regarding clinical relevance, the occlusal plane is the preferred reference plane to identify the ML and the MF for preventing injury to the inferior alveolar neurovascular bundle.”

      9. The limitations of the study need to be added.

Answer: In the last paragraph of Discussion section, we add the “Limitations of the literature review were (1) varied reference planes, (2) rare report in the edentulous people, (3) imbalance in the age distribution of the population, and (4) distribution disequilibria in racial population.”

Reviewer 3 Report

To editor and author

The manuscript described the details of the information. However, I wonder if the author could define the differences between race (biological terminology) and ethnicity  (social terminology) in the discussion. Would the author be able to discuss on the anatomical landmarks being different depending on the social terminology?

Thank you very much

Author Response

Reviewer 3

The manuscript described the details of the information. However, I wonder if the author could define the differences between race (biological terminology) and ethnicity (social terminology) in the discussion. Would the author be able to discuss on the anatomical landmarks being different depending on the social terminology?

Answer: The “ethnicity” is changed to “race”.

  1. In the last sentence of Abstract section,” No significant difference was observed in the location and related distances for the ML among patients of different sexes, races, and skeletal patterns.”
  2. In the last sentence of Introduction section, “This difference may be attributed to age, race, reference plane, skeletal pattern, and the preparation of dry mandibles [9-14].”
  3. In the first paragraph of Discussion section, “The outcomes of eligible studies for different sexes, races, and skeletal patterns are discussed below.”
  4. In the 4.1. ML height, “We found that the ML height was similar in the dry mandible and CBCT groups. They were also similar across different races and skeletal patterns.”
  5. In the 4.7. Clinical relevance and limitations of the study, “In future studies, the sample distribution should consider the age and race of the study population.”
  6. In the Conclusion section, “Differences in age, race, and sex of participants; reference planes; and reference points may have generated different measurement results.”

Round 2

Reviewer 2 Report

The authors have addressed most of the comments however still missing a proper format for table 1, which is completely messed up. Definitely, this has to be improved or split into the main table with all the important demographic parameters and the other ones as a supplementary table or adjust the format for each parameter if you want to provide also the race. 

Author Response

Table 1 and Table 2 are revised to adjust the parameters including race.

Sincerely yours,

Kun-Jung Hsu, DDS, PhD.

Graduate Institute of Dental Sciences, College of Dental Medicine,

Kaohsiung medical university.